

# Psychometric properties of instruments for assessing adherence to oral antineoplastic agents: a COSMIN systematic review

Miaomiao Sun*, Kanghui Huang*, Suxiang Liu, Chuchu Fang and Lili Yang

Department of Nursing, The Fourth Affiliated Hospital of School of Medicine, and International School of Medicine, International Institutes of Medicine, Zhejiang University, Yiwu, China
* These authors contributed equally to this work.

## ABSTRACT

**Introduction.** Oral antineoplastic therapies have increasingly become a mainstay therapy for various cancers. Selecting the most suitable instrument for assessing adherence to oral antineoplastic agents (OAAs) in cancer patients is crucial to tracking patients' medication compliance. This review is the first to identify available instruments for assessing adherence to OAAs and examine the quality of their psychometric properties.

**Methods.** Following the Consensus-based Standards for the Selection of Health Measurement Instruments (COSMIN) guidelines and Preferred Reporting Items for Systematic Reviews and Meta-Analyses (PRISMA) guidelines, eight electronic databases (PubMed, Embase, Web of Science, Cumulative Index to Nursing and Allied Health Literature (CINAHL), China National Knowledge Infrastructure (CNKI), Wanfang, Weipu, and Sinomed) were systematically searched for relevant studies published from inception until December 31, 2023. The study protocol received registration with the International Prospective Register of Systematic Reviews (CRD42024546402).

**Results.** Eight studies assessing eight identified instruments were included. Four instruments were universal to cancer patients treated with any OAA; the other four instruments were only suitable for a specific type of OAA. None of the studies explored measurement error, cross-cultural validity/measurement invariance, and responsiveness of the instruments. All the instruments failed to meet the COSMIN criteria. Eventually, seven instruments were weakly recommended for use to assess OAA adherence, and one was not recommended for assessing OAA adherence due to high-quality evidence for insufficient internal consistency.

**Conclusion.** The selection of the most appropriate instrument to assess adherence to OAAs depends on its psychometric properties and relevance to the type of OAA. Eight identified instruments for assessing adherence to OAAs demonstrated limited reliability and validity. Further thorough validation is required for all included instruments. Instruments with rigid measurement properties are urgently needed to be developed to assess OAA adherence in cancer patients.

Corresponding author
Lili Yang, 3200006@zju.edu.cn

## INTRODUCTION

Since the 1990s, oral antineoplastic therapies have increasingly become a mainstay therapy for various cancers, ranging from traditional endocrine, immunomodulatory, and cytotoxic therapies to drugs targeting genetic mutations (*Bassan et al., 2014*; *Burstein et al., 2014*). With oral antineoplastic agents (OAAs), patients can shorten the length of hospital stays and avoid the pain of intravenous injection (*Borner et al., 2001*). However, patients and their caregivers are the primary managers and implementers of OAAs, as OAAs are mainly administered at home. Thus, poor medication adherence is one of the crucial concerns for those patients (*McCue, Lohr & Pick, 2014*). Medication adherence is the degree to which the person's behavior corresponds with the agreed recommendations from a healthcare provider (*Sabaté, 2003*). Poor medication adherence can impair treatment outcomes, exacerbate adverse effects, increase hospital visits, incur unnecessary escalation to more expensive treatments, and lead to a waste of healthcare resources (*Makubate et al., 2013*; *McCowan et al., 2013*; *Vyas et al., 2024*). Medication adherence to OAAs among cancer patients is multi-factorial, complex, and influenced by patient-related, therapy-related, disease-related, healthcare system, and socioeconomic factors (*Verbrugghe et al., 2013*).

Ensuring medication adherence is the key to ensuring optimal treatment outcomes for cancer patients (*Lasala & Santoleri, 2022*); it becomes essential to have instruments that measure adherence to OAAs accurately and effectively. Current practices for measuring medication adherence in oncology literature include both objective and subjective measures (*Atkinson et al., 2016*). Objective measures include pill counts, medication possession ratio, pharmacy refill rates, and the Medication Event Monitoring System (MEMSCap™), whereas subjective measures consist of various patient-reported outcome measures (PROMs). Some of the objective measures may not be sustainable because they are too costly or not time-effective. In contrast, PROMs are easy to use in clinical settings and allow clinical healthcare professionals and researchers to receive timely feedback and actively improve patients' medication adherence (*Kwan et al., 2020*).

Two studies in China reported adherence rates of OAAs ranging from 29% to 82% (*Chen et al., 2020*; *Chen et al., 2022*). A systematic review reported that rates for adherence to OAAs among cancer patients ranged from 23% to 100% because different measures were used (*Huang et al., 2016*). Most studies used the Morisky Medication Adherence Scale (MMAS-8) (*Morisky, Green & Levine, 1986*) or its modified form to assess OAA adherence. Originally developed for patients with hypertension, the MMAS-8 (*Morisky, Green & Levine, 1986*) applicability and effectiveness in cancer patients treated with OAAs remains to be verified. OAAs are significantly different from other chronic disease medications (*e.g.*, antihypertensive drugs). Firstly, the side effects of OAA are often more varied, pronounced, and severe, which might impair medication adherence. Secondly, OAAs tend to be more expensive, which may deter cancer patients from adhering to treatment due to financial constrains (*Neugut et al., 2011*). Additionally, chemotherapy regimens can be quite complex, for example, having to consume five to 12 pills two to three times daily, and may involve confusing schedules like two weeks on treatment, one week off, and then two more weeks on (*Baudot et al., 2016*). However, the above-discussed factors are not
emphasized in the non-specific OAA adherence scales. Therefore, it is essential to utilize specific OAA adherence scales to better identify the factors that may hinder or favor cancer patients' medication adherence.

Selecting the best PROM among the many available for measuring OAA adherence in cancer patients is challenging. The Consensus-based Standards for the Selection of Health Measurement Instruments (COSMIN) initiative published a guideline for conducting systematic reviews of studies evaluating the measurement properties of PROMs (*Prinsen et al., 2018*). This guideline (*Prinsen et al., 2018*) proposed the criteria to assess the methodological quality of studies on measurement properties and the quality of the self-reported measurement, which helps researchers choose the best PROM.

Previously, a systematic review (*Claros, Messa & García-Perdomo, 2019*) of instruments for oral pharmacological treatment adherence in cancer patients was conducted. However, it has the following disadvantages. On the one hand, it targeted oral therapy that included not only OAAs but also other oral drugs such as analgesics. Since the side effects of OAAs are usually more severe than those of analgesics, adherence to OAAs and analgesics varies substantially in cancer patients (*Shrestha et al., 2024*). On the other hand, it did not use the COSMIN guideline (*Prinsen et al., 2018*) to evaluate PROMs. That review (*Claros, Messa & García-Perdomo, 2019*) assessed the risk of bias for each included study following the adaptation of the Strengthening the Reporting of Observational Studies in Epidemiology (STROBE) guidelines (*Vandenbroucke et al., 2007*). It is important to note that STROBE (*Vandenbroucke et al., 2007*) is not a specific tool for evaluating the methodological quality of measurement properties of PROMs. While the COSMIN guideline (*Prinsen et al., 2018*) established "standards" and "criteria" for measurement properties of PROMs.

No systematic review has summarized all the instruments of adherence to OAAs and reported the quality of their psychometric properties. Therefore, this study aims to adopt the COSMIN guideline (*Prinsen et al., 2018*) to critically assess, compare, and synthesize the quality of the psychometric properties of PROMs for OAA adherence in adult cancer patients. Our attempts may provide evidence and a reference for selecting appropriate adherence instruments to assess cancer patients' OAA adherence.

## METHODS

This systematic review followed the COSMIN guideline for conducting systematic reviews of PROMs (*Prinsen et al., 2018*) and was reported following the Preferred Reporting Items for Systematic Reviews and Meta-Analyses (PRISMA) statement (Table S1) (*Page et al., 2021*). The study protocol received registration with the International Prospective Register of Systematic Reviews (CRD42024546402).

### Search strategy
Four English databases (PubMed, EMBASE, Web of Science, and Cumulative Index to Nursing and Allied Health Literature (CINAHL)) and four Chinese databases (China National Knowledge Infrastructure (CNKI), Wanfang, Weipu, and Sinomed) were searched for relevant studies published from the database inception to December 31, 2023. A search strategy (Table S2) consisting of medication adherence, cancers, oral administration,

PROMs, and measurement properties was used. The search filters (*Terwee et al., 2009*) were also used to enhance the sensitivity of searches, where available. Additionally, all retrieved papers' reference lists were manually searched for potential studies.

## Inclusion and exclusion criteria

The inclusion criteria were as follows: (1) studies that focused on the adherence to OAAs, including endocrine, cytotoxic, targeted therapies and immunomodulators across cancer types; (2) the target population consisted of adult patients (age > 18 years) diagnosed with cancer; (3) studies aiming to develop or validate the PROMs; (4) studies describing at least one psychometric property listed in the COSMIN; (5) full-text articles are available.

The exclusion criteria included the following: (1) studies that included mainly oral agents other than OAAs (such as analgesics and antiemetics); (2) studies in which a PROM was solely used to measure an outcome (*e.g.*, randomized clinical trials); (3) repeated or overlapping publication.

## Study selection

We imported all the references searched and removed duplicates in EndNote X9. The formal screening was conducted in April 2024. Two reviewers (MMS, KHH), trained in evidence-based methodologies, independently screened the titles and abstracts of the references according to the predefined eligibility criteria and then retrieved the full text of articles that may have wholly or potentially met the eligibility criteria. A third reviewer (SXL) resolved disagreements.

## Data extraction

Two reviewers (MMS, KHH) performed the data extraction independently in June 2024. A third reviewer (SXL) resolved disagreements. A standardized extraction table was utilized, which included the details of the studies (first author, published year, country or region, the PROM used, participants' characteristics, study design, response rate, and article's language), characteristics of the included PROMs (item generation, dimensions and items, target population, range of scores, mode of administration, response options, original language/translation, and measurement properties). Original authors were contacted for missing or unreported data.

## Appraisal of methodological quality and measurement properties

Two reviewers (MMS, KHH) independently appraised the methodological quality and measurement properties of the PROMs. A third reviewer (SXL) resolved disagreements.

The methodological quality of the eligible studies was rated using the COSMIN Risk of Bias checklist (*Mokkink et al., 2018*). The checklist consists of ten boxes: PROM development, content validity, structural validity, internal consistency, cross-cultural validity/measurement invariance, reliability (test–retest), measurement error, criterion validity, hypothesis testing for construct validity, and responsiveness. The methodological quality of items in each box was rated as "very good," "adequate," "doubtful," or "inadequate." The quality of the criteria for good measurement properties of each PROM was rated as sufficient (+), insufficient (-), inconsistent (±), or indeterminate (?), according to the proposed criteria (*Mokkink et al., 2018*; *Prinsen et al., 2018*).

Since the COSMIN does not provide evaluation criteria for assessing structural validity using exploratory factor analysis (EFA), we considered a sufficient result when the total variance explained was at least 60% (*Wu, 2010*). In addition, we suggested that the statistical results from assessing the relationship between the PROM and the direct objective measures should be viewed as a criterion validity result when evaluating criterion validity (*Oliveira et al., 2023*).

## Grading the quality of evidence and formulate recommendations

Two reviewers (MMS, KHH) independently graded the quality of the evidence of all studies according to the adapted version of the Grading of Recommendations Assessment, Development, and Evaluation (GRADE), proposed by the COSMIN initiative (*Guyatt et al., 2011*; *Prinsen et al., 2018*). A third reviewer (SXL) resolved disagreements. The quality of evidence for each measurement property was graded as "high", "moderate", "low", or "very low". COSMIN recommends categorizing the included PROMs into three categories (*Prinsen et al., 2018*): PROMs with evidence for sufficient content validity (at any level) and at least low-quality evidence for sufficient internal consistency can be categorized as "strongly recommended"; PROMs with high-quality evidence for an insufficient psychometric property should be categorized as "not recommended"; and other situations are categorized as "weakly recommended."

## RESULTS

### Literature search

A total of 1,985 records were initially identified from the databases. After removing duplicates, title/abstract screening, full-text reviews, and additional searches, eight studies and eight instruments were eventually included in the systematic review (see Fig. 1).

### Characteristics of included study and PROMs

Table 1 presents detailed information on the studies that were included. Eight studies were published between 2013 and 2023. There were five studies reported in English (*Bagcivan & Akbayrak, 2015*; *Daouphars et al., 2013*; *Gambalunga et al., 2022*; *Silveira et al., 2021*; *Talens et al., 2023*), two in Chinese (*Li, Sun & Dong, 2018*; *Qin et al., 2020*), and one in French (*Baudot et al., 2016*). France (*Baudot et al., 2016*; *Daouphars et al., 2013*) and China (*Li, Sun & Dong, 2018*; *Qin et al., 2020*) each contributed two papers; Turkey (*Bagcivan & Akbayrak, 2015*), Brazil (*Silveira et al., 2021*), Italy (*Gambalunga et al., 2022*) and Spain (*Talens et al., 2023*) each published one paper. Except for two cohort studies (*Baudot et al., 2016*; *Daouphars et al., 2013*), the remaining studies were cross-sectional (*Bagcivan & Akbayrak, 2015*; *Gambalunga et al., 2022*; *Li, Sun & Dong, 2018*; *Qin et al., 2020*; *Silveira et al., 2021*; *Talens et al., 2023*). The sample sizes ranged from 46 to 306. The mean/median age ranged from 54.6 to 68.6 years. Most of the cancer diagnoses of the participants were breast cancer, gastrointestinal tract cancer, hematologic cancer, lung cancer, and genitourinary system cancer. The OAAs taken by the participants mainly included cytotoxic drugs (*e.g.*, capecitabine, tegafur, etoposide, hydroxyurea, and vinorelbine), targeted drugs (*e.g.*, imatinib, sunitinib, and ibrutinib), hormone drugs (*e.g.*,

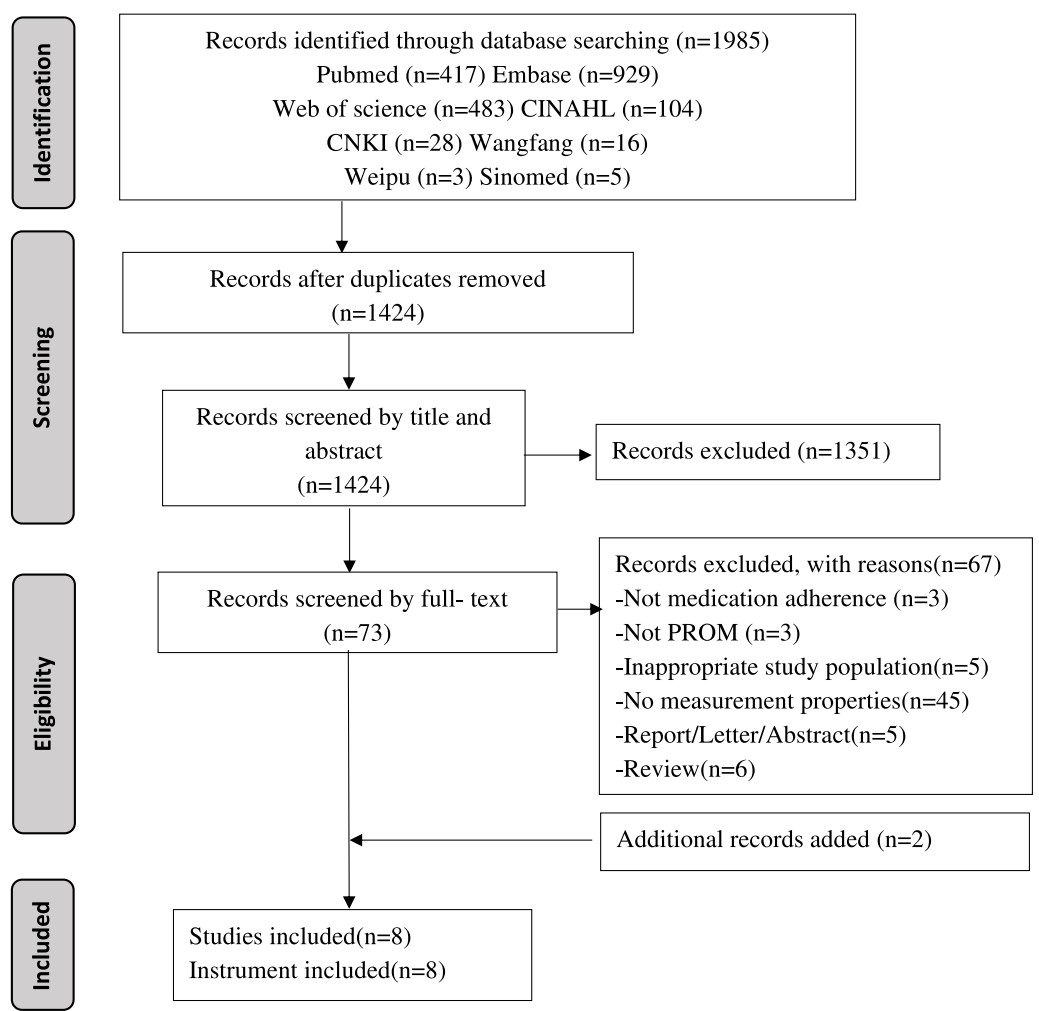

**Figure 1** **Flowchart of literature search and study selection.**

tamoxifen, anastrozole, exemestane, letrozole, and abiraterone) and immunomodulator (*e.g.*, lenalidomide, thalidomide, and pomalidomide). Three studies (*Daouphars et al., 2013*; *Li, Sun & Dong, 2018*; *Talens et al., 2023*) reported a PROM response rate ranging from 93.9% to 98.0%.

Table 2 describes detailed information on the included PROMs. Eight PROMs were identified, four of which were universal to cancer patients treated with any OAA: the Turkish-Version Oral Chemotherapy Adherence Scale (T-OCAS) (*Bagcivan & Akbayrak, 2015*), Chinese-Version Oral Chemotherapy Adherence Scale (C-OCAS) (*Li, Sun & Dong, 2018*), Morisky Medication Adherence Scale-8 (MMAS-8) (*Qin et al., 2020*), and Experience with and Adherence to Oral Antineoplastic Agents Scale (EXPAD-ANEO) (*Talens et al., 2023*). The other four instruments were only suitable for patients with specific types of cancer or patients using a particular kind of OAA: the Self-assessment Adherence Questionnaire (*Daouphars et al., 2013*) for patients with chronic myeloid

**Table 1  Study characteristics.**

| Authors (year) | PROM | Study design | Participants | | | | Response rate | Country | Language |
|---|---|---|---|---|---|---|---|---|---|
| | | | N | Age | Cancer Diagnosis | OAAs taken | | | |
| *Daouphars et al. (2013)* | Self-assessment Adherence Questionnaire | Retrospective cohort study | 46 | 59.0[a] | Chronic myeloid leukemia | Imatinib | 93.9% | France | English |
| *Bagcivan & Akbayrak (2015)* | Turkish-Version Oral Chemotherapy Adherence Scale (T-OCAS) | Cross-sectional study | 306 | 68.6[a] | Breast cancer, gastrointestinal tract cancer, lung cancer, genitourinary system cancer, hematologic cancer, etc | Capecitabine, imatinib, sunitinib, etoposide, hydroxyurea, vinorelbine et al. | NR | Turkey | English |
| *Baudot et al. (2016)* | Measuring Adherence and Management of Side Effects in Patients Treated with Capecitabine | Prospective cohort study | 67 | 54.6[a] | Breast cancer, colon cancer | Capecitabine | NR | France | French |
| *Li, Sun & Dong (2018)* | Chinese-Version Oral Chemotherapy Adherence Scale (C-OCAS) | Cross-sectional study | 201 | NR | Not specific | Not specific | 98.0% | China | Chinese |
| *Qin et al. (2020)* | Morisky Medication Adherence Scale-8 (MMAS-8) | Cross-sectional study | 75 | 66.0[m] | Gastric cancer, colorectal cancer | Tegafur or capecitabine | NR | China | Chinese |
| *Silveira et al. (2021)* | Treatment Adherence Measure (TAM) | Cross-sectional study | 84 | 62.7[m] | Multiple myeloma | Thalidomide, lenalidomide, pomalidomide | NR | Brazil | English |
| *Gambalunga et al. (2022)* | Adherence-Breast Endocrine Therapy Questionnaire (A-BET) | Cross-sectional study | 82 | 56.4[a] | Breast cancer | Tamoxifen, anastrazole, exemestane, letrozole | NR | Italy | English |
| *Talens et al. (2023)* | Experience with and Adherence to Oral Antineoplastic Agents Scale (EXPAD-ANEO) | Cross-sectional study | 268 | 64.1[a] | Multiple myeloma, breast cancer, chronic lymphocytic leukemia, prostate cancer | Lenalidomide, ibrutinib, capecitabine, abiraterone | 95.0% | Spain | English |

**Notes.**

NR, not reported; OAAs, oral antineoplastic agents; a, means the average of age; m, means the median of age.

**Table 2  Characteristics of the included PROMs.**

| Authors (year) | PROM | Item generation | Dimensions and items | Target population | Mode of administration | Response options | Range of scores | Original language/ translation |
|---|---|---|---|---|---|---|---|---|
| *Daouphars et al. (2013)* | Self-Assessment Adherence Questionnaire | Modeled after previously published questionnaires related to other disease states | 10 items | Patients with chronic myeloid leukemia treated with imatinib | self-report | Two choices: "yes" or "no" | 0–10 | French |
| *Bagcivan & Akbayrak (2015)* | Turkish-Version Oral Chemotherapy Adherence Scale (T-OCAS) | Based on Cox's Interaction Model of Client Health Behavior (IMCHB), literature review and Delphi expert consultation | 19 items divided into 3 dimensions: expected behaviors related to the treatment period, barriers, and expected behaviors during drug use | Patients taking OAAs | Interview-based | Likert-5: 1=never; 5=always | 1–95 | Turkish |
| *Baudot et al. (2016)* | Measuring Adherence and Management of Side Effects in Patients Treated with Capecitabine | Based on previous qualitative research and a multidisciplinary team | 6 items | Patients treated with capecitabine | self-report | Proposed answers for each scenario | 0–120 | French |
| *Li, Sun & Dong (2018)* | Chinese-Version Oral Chemotherapy Adherence Scale (C-OCAS) | Chinese translation and cross-cultural adaptation of the original scale (T-OCAS) (*Bagcivan & Akbayrak, 2015*) | 16 items divided into 3 dimensions: expected behaviors related to the treatment period, barriers, and expected behaviors during drug use | Patients taking OAAs | self-report | Likert-5: 1 = never; 5 = always | 1–80 | Turkish/Chinese |
| *Qin et al. (2020)* | Morisky Medication Adherence Scale-8 (MMAS-8) | Based on the Chinese version of MMAS-8 that has been translated and cross-cultural adapted by other Chinese researchers (*Wang, Mo & Bian, 2013*) | 8 items divided into 3 dimensions: various conditions that cause forgetting to take medications, confidence in adherence to the treatment plan, and difficulty in taking medications accurately | Patients taking OAAs | self-report | Two choices: "yes" or "no" | 0–8 | English/Chinese |
| *Silveira et al. (2021)* | Treatment Adherence Measure (TAM) | Based on the Brazilian Portuguese version of TAM that has been cross-cultural adapted by other Brazilian researchers (*Borba et al., 2018*) | 7 items | Patients with multiple myeloma treated with immunomodulators | Interview-based | Likert-6: 1 = always; 6 = never | 1–6 | Portuguese/ Brazilian Portuguese |
| *Gambalunga et al. (2022)* | Adherence-Breast Endocrine Therapy Questionnaire (A-BET) | Based on the literature review and an analysis of the related questionnaires available, previous qualitative research, and a focus group of experts | 6 items | Patients with breast cancer treated with oral endocrine therapy | Interview-based | Inconsistent options | NR | Italian |
| *Talens et al. (2023)* | Experience with and Adherence to Oral Antineoplastic Agents Scale (EXPAD-ANEO) | Based on the literature review, previous qualitative research, and a steering committee | 7 items divided into 2 dimensions: beliefs and expectations about treatment, behavior, and attitudes | Patients taking OAAs | Interview-based | Two choices: "yes" or "no" | 0–7 | Spanish |

**Notes.**

NR,  not reported;  OAAs,  oral antineoplastic agents.

leukemia treated with imatinib; Measuring Adherence and Management of Side Effects in Patients Treated with Capecitabine (*Baudot et al., 2016*) for patients treated with capecitabine; Treatment Adherence Measure (TAM) (*Silveira et al., 2021*) for patients with multiple myeloma treated with immunomodulators; Adherence-Breast Endocrine Therapy Questionnaire (A-BET) (*Gambalunga et al., 2022*) for patients with breast cancer treated with oral endocrine therapy. Although all identified instruments were validated among cancer patients taking OAAs, three PROMs (*Daouphars et al., 2013*; *Qin et al., 2020*; *Silveira et al., 2021*) were not initially developed for cancer patients taking OAAs: MMAS-8 was developed for patients with essential hypertension (*Morisky, Green & Levine, 1986*); TAM was designed for patients with chronic diseases (*Delgado & Lima, 2001*); and the Self-assessment Adherence Questionnaire was modeled after previously published questionnaires related to other disease states (*Chisholm et al., 2005*; *Girerd et al., 2001*). One PROM has Turkish (*Bagcivan & Akbayrak, 2015*) and Chinese versions (*Li, Sun & Dong, 2018*), which we treated as independent instruments because their items and subscale compositions varied. All instruments were either self-reported by patients through self-completed questionnaires or interview-based approaches. The summary of measurement properties for each PROM is presented in Table S3.

## Methodological quality of included studies

Table 3 demonstrates the methodological quality results of the included studies. Five studies presented the PROM development process (*Bagcivan & Akbayrak, 2015*; *Baudot et al., 2016*; *Daouphars et al., 2013*; *Gambalunga et al., 2022*; *Talens et al., 2023*). The quality of development studies for two PROMs (*Bagcivan & Akbayrak, 2015*; *Daouphars et al., 2013*) was rated inadequate as no appropriate qualitative methods were used to evaluate relevant items for the instruments. The quality of three PROM development studies (*Baudot et al., 2016*; *Gambalunga et al., 2022*; *Talens et al., 2023*) was rated doubtful because each item was tested in an inappropriate number of patients or it was unclear if all items were tested in their final form. Five studies (*Bagcivan & Akbayrak, 2015*; *Baudot et al., 2016*; *Gambalunga et al., 2022*; *Li, Sun & Dong, 2018*; *Talens et al., 2023*) reported content validity, and all of them were rated adequate or doubtful because it was unclear if professionals from all required disciplines were included or each item was tested in an inappropriate number of patients or professionals.

For structural validity, one study (*Talens et al., 2023*) was rated very good owing to using confirmatory factor analysis (CFA), three studies (*Bagcivan & Akbayrak, 2015*; *Li, Sun & Dong, 2018*; *Qin et al., 2020*) were rated adequate with EFA conducted, and one study (*Baudot et al., 2016*) was rated inadequate with correlation coefficient reported. All included studies assessed internal consistency. Two studies (*Li, Sun & Dong, 2018*; *Talens et al., 2023*) were rated very good due to calculating Cronbach's alpha or omega for each subscale. Three studies (*Daouphars et al., 2013*; *Gambalunga et al., 2022*; *Silveira et al., 2021*) were rated doubtful due to uncertainty if the scale was unidimensional. Three studies (*Bagcivan & Akbayrak, 2015*; *Baudot et al., 2016*; *Qin et al., 2020*) were rated inadequate because no Cronbach's alpha was calculated for each internal subscale, or only the correlation coefficient between entries was calculated.

Sun et al. (2025), *PeerJ*, DOI 10.7717/peerj.19088

**Table 3  Methodological quality assessment.**

| PROM | Authors (year) | Development | Content validity | | | Structural validity | Internal consistency | Reliability | Criterion validity | Hypotheses testing |
|---|---|---|---|---|---|---|---|---|---|---|
| | | | Relevance | Comprehensiveness | Comprehensibility | | | | | |
| Self-assessment adherence questionnaire | *Daouphars et al. (2013)* | I | | | | | D | | I | |
| T-OCAS | *Bagcivan & Akbayrak (2015)* | I | A[b] | | A[a] | A | I | D | I | |
| Measuring Adherence and Management of Side Effects in Patients Treated with Capecitabine | *Baudot et al. (2016)* | D | D[ab] | D[ab] | D[a] | I | I | | | V |
| C-OCAS | *Li, Sun & Dong (2018)* | | D[b] | | | A | V | D | | |
| MMAS-8 | *Qin et al. (2020)* | | | | | A | I | D | | |
| TAM | *Silveira et al. (2021)* | | | | | | D | | | V |
| A-BET | *Gambalunga et al. (2022)* | D | A[ab] | | A[a] | | D | D | | |
| EXPAD-ANEO | *Talens et al. (2023)* | D | D[ab] | D[ab] | D[a] | V | V | | I | A |

**Notes.**

Empty cell means no reported testing; To determine the overall rating of the quality of each single study on a measurement property, the lowest rating of any standard in the box is taken (*i.e.*, "the worst score counts" principle): V, very good; A, adequate; D, doubtful; I, inadequate; a, means asking patients; b, means asking professionals.

T-OCAS, Turkish-Version Oral Chemotherapy Adherence Scale; C-OCAS, Chinese-Version Oral Chemotherapy Adherence Scale; MMAS-8, Morisky Medication Adherence Scale-8; TAM, Treatment Adherence Measure; A-BET, Adherence-Breast Endocrine Therapy Questionnaire; EXPAD-ANEO, Experience with and Adherence to Oral Antineoplastic Agents Scale.

Four studies (*Bagcivan & Akbayrak, 2015*; *Gambalunga et al., 2022*; *Li, Sun & Dong, 2018*; *Qin et al., 2020*) reported reliability, but none calculated the intraclass correlation coefficient (ICC). Furthermore, none of the four studies specified whether the patients remained stable, if the conditions were comparable, or if the time interval was suitable, making the reliability doubtful. Criterion validity evaluated by three studies (*Bagcivan & Akbayrak, 2015*; *Daouphars et al., 2013*; *Talens et al., 2023*) was rated inadequate due to lack of the area under the receiver operating curve (AUC) or choosing another medication adherence tool for comparison. Hypothesis testing for construct validity assessed by three studies (*Baudot et al., 2016*; *Silveira et al., 2021*; *Talens et al., 2023*) was classified as very good or adequate. None of the PROMs explored measurement error, cross-cultural validity/measurement invariance, and responsiveness.

## Quality of psychometric properties and evidence

Table 4 displays the quality of psychometric properties and evidence for each PROM. Two PROMs (*Baudot et al., 2016*; *Talens et al., 2023*) demonstrated sufficient quality for content validity. The quality of content validity of three PROMs (*Bagcivan & Akbayrak, 2015*; *Gambalunga et al., 2022*; *Li, Sun & Dong, 2018*) was rated indeterminate due to a lack of asking the patients about the relevance of the items or the patients and professionals about the comprehensiveness of the items. Two PROMs (*Bagcivan & Akbayrak, 2015*; *Gambalunga et al., 2022*) had high-quality evidence of content validity, three PROMs (*Baudot et al., 2016*; *Li, Sun & Dong, 2018*; *Talens et al., 2023*) had moderate-quality evidence, and one PROM (*Daouphars et al., 2013*) had very low-quality evidence owing to no content validity studies and the inadequate quality of the PROM development study.

Three PROMs (*Li, Sun & Dong, 2018*; *Qin et al., 2020*; *Talens et al., 2023*) had sufficient quality for structural validity, one PROM (*Baudot et al., 2016*) had indeterminate quality, and one PROM (*Bagcivan & Akbayrak, 2015*) had insufficient quality because the total variance explained was merely 43%. The quality of evidence for structural validity of one PROM (*Talens et al., 2023*) was high, whereas three PROMs (*Bagcivan & Akbayrak, 2015*; *Li, Sun & Dong, 2018*; *Qin et al., 2020*) were downgraded to moderate quality of evidence due to a serious risk of bias, and one PROM (*Baudot et al., 2016*) was downgraded to very low due to an extremely serious risk of bias.

One PROM (*Talens et al., 2023*) had insufficient internal consistency because the Omega coefficient was less than 0.7 for one of the factors. One PROM's (*Li, Sun & Dong, 2018*) internal consistency was rated sufficient; the other six PROMS (*Bagcivan & Akbayrak, 2015*; *Baudot et al., 2016*; *Daouphars et al., 2013*; *Gambalunga et al., 2022*; *Qin et al., 2020*; *Silveira et al., 2021*) were rated indeterminate because the criteria for "at least low evidence for sufficient structural validity" were not met, or Cronbach's alpha for each subscale was not reported. The quality of evidence for internal consistency of two PROMs (*Li, Sun & Dong, 2018*; *Talens et al., 2023*) was high, one PROM (*Bagcivan & Akbayrak, 2015*) was downgraded to low due to a very serious risk of bias, and five PROMs (*Baudot et al., 2016*; *Daouphars et al., 2013*; *Gambalunga et al., 2022*; *Qin et al., 2020*; *Silveira et al., 2021*) were downgraded to very low due to a very serious risk of bias and imprecision.

## Table 4 The quality of psychometric property measurement and evidence.

| PROM | Authors (year) | Content validity | | Structural validity | | Internal consistency | | Reliability | | Criterion validity | | Hypotheses testing | | Recommendation for use to assess adherence to OAA |
|---|---|---|---|---|---|---|---|---|---|---|---|---|---|---|
| | | QM | QE | QM | QE | QM | QE | QM | QE | QM | QE | QM | QE | |
| Self-Assessment Adherence Questionnaire | Daouphars et al. (2013) | | Very low | | | ? | Very low | | | ? | Very low | | | Weakly recommended |
| T-OCAS | Bagcivan & Akbayrak (2015) | ? | High | – | Moderate | ? | Low | ? | Low | ? | Very low | | | Weakly recommended |
| Measuring Adherence and Management of Side Effects in Patients Treated with Capecitabine | Baudot et al. (2016) | + | Moderate | ? | Very low | ? | Very low | | | | | + | Moderate | Weakly recommended |
| C-OCAS | Li, Sun & Dong (2018) | ? | Moderate | + | Moderate | + | High | ? | Very low | | | | | Weakly recommended |
| MMAS-8 | Qin et al. (2020) | | | + | Moderate | ? | Very low | ? | Very low | | | | | Weakly recommended |
| TAM | Silveira et al. (2021) | | | | | ? | Very low | | | | | – | Moderate | Weakly recommended |
| A-BET | Gambalunga et al. (2022) | ? | High | | | ? | Very low | ? | Very low | | | | | Weakly recommended |
| EXPAD-ANEO | Talens et al. (2023) | + | Moderate | + | High | – | High | | | ? | Low | + | Moderate | Not recommended |

**Notes.**

Empty cell means no reported testing; QM, quality of measurement; QE, quality of evidence; +, sufficient; ?, indeterminate; -, insufficient; COSMIN recommends categorizing the included PROMs into three categories (*Prinsen et al., 2018*): strongly recommended = PROMs with evidence for sufficient content validity (at any level) and at least low-quality evidence for sufficient internal consistency, not recommended = PROMs with high-quality evidence for an insufficient psychometric property, weakly recommended = other situations.

T-OCAS, Turkish-Version Oral Chemotherapy Adherence Scale; C-OCAS, Chinese-Version Oral Chemotherapy Adherence Scale; MMAS-8, Morisky Medication Adherence Scale-8; TAM, Treatment Adherence Measure; A-BET, Adherence-Breast Endocrine Therapy Questionnaire; EXPAD-ANEO, Experience with and Adherence to Oral Antineoplastic Agents Scale.

Four PROMs (*Bagcivan & Akbayrak, 2015*; *Gambalunga et al., 2022*; *Li, Sun & Dong, 2018*; *Qin et al., 2020*) failed to report ICC, leaving indeterminate quality for reliability. The quality of evidence for the reliability of one PROM (*Bagcivan & Akbayrak, 2015*) was downgraded to low due to a very serious risk of bias, and three PROMs (*Gambalunga et al., 2022*; *Li, Sun & Dong, 2018*; *Qin et al., 2020*) were downgraded to very low due to a very serious risk of bias and imprecision. Three PROMs (*Bagcivan & Akbayrak, 2015*; *Daouphars et al., 2013*; *Talens et al., 2023*) reported indeterminate quality for criterion validity owing to inappropriate selection of the gold standard or the failure to calculate AUC. The quality of evidence for the criterion validity of one PROM was downgraded to low due to a very serious risk of bias (*Talens et al., 2023*), and two PROMs (*Bagcivan & Akbayrak, 2015*; *Daouphars et al., 2013*) were downgraded to very low due to a very serious risk of bias and imprecision. The quality of hypothesis testing for construct validity for two PROMs (*Baudot et al., 2016*; *Talens et al., 2023*) was sufficient; one PROM (*Silveira et al., 2021*) was insufficient because only 50% of the hypothesis was confirmed. The quality of evidence for hypothesis testing for construct validity for three PROMs (*Baudot et al., 2016*; *Silveira et al., 2021*; *Talens et al., 2023*) was downgraded to moderate due to a serious risk of bias or a sample size of less than 100.

### Recommended grade of PROMs

The results of the recommended grades for eight PROMs are shown in Table 4. Seven PROMs (*Bagcivan & Akbayrak, 2015*; *Baudot et al., 2016*; *Daouphars et al., 2013*; *Gambalunga et al., 2022*; *Li, Sun & Dong, 2018*; *Qin et al., 2020*; *Silveira et al., 2021*) were weakly recommended for use to assess OAA adherence, whereas one PROM (*Talens et al., 2023*) was not recommended for use to assess OAA adherence due to high-quality evidence for insufficient internal consistency.

## DISCUSSION

This review identified eight PROMs among adult cancer patients to assess adherence to OAAs, including endocrine, cytotoxic, targeted therapies, and immunomodulators. However, none met the COSMIN criteria, and at best were only weakly recommended for assessing OAA adherence. In addition, none of the studies explored measurement error, cross-cultural validity/measurement invariance, and responsiveness of the instruments.

According to the COSMIN guidelines (*Prinsen et al., 2018*), content validity is considered the most critical psychometric property of PROMs. Failure to obtain high content validity may affect all other measurement properties (*Terwee et al., 2018*). However, only five PROMs in this review assessed content validity (*Bagcivan & Akbayrak, 2015*; *Baudot et al., 2016*; *Gambalunga et al., 2022*; *Li, Sun & Dong, 2018*; *Talens et al., 2023*). The other three: MMAS-8 (*Qin et al., 2020*), TAM (*Silveira et al., 2021*), and the self-assessment adherence questionnaire (*Daouphars et al., 2013*) were not originally developed for cancer patients. Disappointingly, none of these PROMs (*Daouphars et al., 2013*; *Qin et al., 2020*; *Silveira et al., 2021*) assessed content validity when applied to cancer patients treated with OAAs. Thus, content validity must be reassessed when applying the scale to a new target

population, and the target population should not be neglected when asking about the relevance, comprehensiveness, and comprehensibility of scale items (*Terwee et al., 2018*).

The COSMIN checklist (*Mokkink et al., 2018*) suggests that CFA is preferable to EFA, and only the criteria of CFA are available for good quality structural validity (*Prinsen et al., 2018*). However, three studies (*Bagcivan & Akbayrak, 2015*; *Li, Sun & Dong, 2018*; *Qin et al., 2020*) used EFA, and the criteria for EFA are not provided in the COSMIN criteria. Hence, our research team agreed that a minimum of 60% variance explained is sufficient when using EFA (*Wu, 2010*). We encourage researchers to adopt CFA to explore structural validity in the future. All studies evaluated internal consistency, but the majority had doubtful or inadequate methodological quality. Therefore, it is crucial to determine the Cronbach's alpha or omega for each subscale, compared to an internal consistency statistic for the overall tool.

TAM (*Silveira et al., 2021*), MMAS-8 (*Qin et al., 2020*), and OCAS (*Li, Sun & Dong, 2018*) are used internationally, and their translated versions have undergone cross-cultural or linguistic adaptation (*Borba et al., 2018*; *Wang, Mo & Bian, 2013*). However, none of these instruments (*Li, Sun & Dong, 2018*; *Qin et al., 2020*; *Silveira et al., 2021*) has been tested for cross-cultural validity to ensure that the instrument is consistent across different versions and to conduct valid and interpretable score comparisons (*Nuevo et al., 2009*). The number of items in the two versions of OCAS is different. The original Turkish version (*Bagcivan & Akbayrak, 2015*) contained 19 items, whereas the translated Chinese version (*Li, Sun & Dong, 2018*) was modified to include 16 items based on correlation and factor analysis; this may also be related to the poor structural validity of the original version of OCAS (*Bagcivan & Akbayrak, 2015*), with an explained variance of only 43%. Future research is urgently warranted to test the cross-cultural validity of instruments to assess OAA adherence in diverse contexts. We also recommend that before cross-culturally adapting an instrument, researchers should ascertain that its original version has sufficient psychometric properties.

Further validation is urgently required for all included instruments. We recommend that future research stress the completeness and precision of validation and analysis methods for all psychometric properties. At the same time, sample size is critical to improving the quality grade of evidence. Researchers should note that the sample size should be at least 100 or more, based on the premise that the sample size is seven times the number of items so as not to reduce the quality grade of the evidence (*Prinsen et al., 2018*).

This review, while comprehensive, had several limitations that highlight the need for more extensive research. Firstly, each included instrument was supported by only one study, which may have resulted in some bias. Secondly, two additional studies (*Daouphars et al., 2013*; *Silveira et al., 2021*) were identified by a manual search of references of the included studies, which may have been due to inadequate search filters and search strategy. Thirdly, this study included one article (*Baudot et al., 2016*) in French. However, reviewers are not familiar with French. To address this, reviewers initially translated the entire French article into English using machine translation and then extracted data from the English version. In addition, someone fluent in both French and English, with expertise in the reliability and validity of the questionnaire, was asked to verify the accuracy of our data extraction.

Fourthly, reviewers identified the latest OAA's adherence assessment tool (*Signorelli et al., 2024*), but unfortunately, it was not included as it has just completed its development stage, and no measurement properties have been assessed. Finally, patient-reported adherence rates were found to be higher than objective measures across studies, indicating an overestimation of patients' true adherence rates by subjective assessment (*Atkinson et al., 2016*). Thus, subjective instruments are no substitute for objective measurements. These limitations underscore the need for more comprehensive and rigorous research in this area.

## CONCLUSIONS

This systematic review provides a comprehensive overview of the quality of measurement properties and methodological quality of OAA adherence instruments for adult cancer patients. In conclusion, we recommend that selecting the most appropriate instrument depends on its psychometric properties and relevance to the type of OAA. Eight identified instruments for assessing adherence to OAAs demonstrated limited reliability and validity. Therefore, further thorough validation is required for all included instruments. Instruments with rigid measurement properties are urgently needed to be developed to assess OAA adherence in cancer patients.

### Funding
The authors received no funding for this work.

### Competing Interests
The authors declare there are no competing interests.

### Author Contributions
- Miaomiao Sun conceived and designed the experiments, performed the experiments, analyzed the data, prepared figures and/or tables, authored or reviewed drafts of the article, and approved the final draft.
- Kanghui Huang conceived and designed the experiments, performed the experiments, analyzed the data, prepared figures and/or tables, authored or reviewed drafts of the article, and approved the final draft.
- Suxiang Liu analyzed the data, authored or reviewed drafts of the article, and approved the final draft.
- Chuchu Fang conceived and designed the experiments, authored or reviewed drafts of the article, and approved the final draft.
- Lili Yang conceived and designed the experiments, authored or reviewed drafts of the article, and approved the final draft.

### Data Availability
    This is a systematic review/meta-analysis.

## Supplemental Information

Supplemental information for this article can be found online at http://dx.doi.org/10.7717/peerj.19088#supplemental-information.

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
