# Peer review of "Psychometric properties of instruments for assessing adherence to oral antineoplastic agents: a COSMIN systematic review"

_PeerJ, doi:10.7717/peerj.19088_

## Round 0.1 · original submission · Minor Revisions

With both reviews at hand, I am pleased to inform you that your manuscript has been accepted with minor revisions.

Both reviewers provided positive feedback on your work while offering constructive comments to further improve the manuscript. However, I agree with your focus on COSMIN guidelines and do not find it necessary to delve into alternative adherence measurement frameworks, as the scope of your systematic review is clearly defined by COSMIN.

In addition to their suggestions, I would recommend using the same standards for Omega as for Alpha. There is no compelling reason for these standards to differ, as Alpha is effectively a special case of Omega, where the lambdas (i.e., factor loadings) are assumed equal. The COSMIN guidelines may not explicitly report standards for Omega because they likely deemed it an unnecessary specification.

Reviewer 1 ·

Basic reporting

Title & Abstract
Title:
The title accurately captures the manuscript’s content and focus.
Abstract:
The abstract effectively summarizes the research's purpose, methodology, and key findings, clearly highlighting the focus on adherence measurement tools for oral antineoplastic agents (OAAs) using the COSMIN framework.

Introduction
• The article is clear unambiguous and professional English is used. The article sets a solid theoretical foundation, explaining the importance of adherence to OAAs in cancer treatment outcomes and the necessity for reliable, validated measurement tools. The review references the COSMIN framework, supporting the need for a systematic review to identify, evaluate, and compare these instruments. This theoretical grounding is relevant and aligns with the stated aims.
• However, we suggest that the article will benefit from adding references on alternative adherence measurement frameworks, beyond COSMIN that could provide a comparative perspective and strengthen the discussion of each tool’s robustness: Citing additional literature on PROMs specifically validated in oncology contexts could reinforce the findings about the limitations of non-specific adherence tools (e.g., Morisky Medication Adherence Scale which is widely used for general medication adherence).
• Briefly outlining the study’s criteria for including tools would add clarity. For example, it should be clear that only studies evaluating psychometric properties of tools specific to OAAs were included.
• Additionally, it would be helpful to mention that adult cancer patients were the focus, excluding studies involving other medications or non-cancer populations.
• Also, including references that compare the effectiveness and reliability of self-reported adherence versus objective metrics would highlight the potential limitations of subjective tools and inform future tool development recommendations.
• In addition, since several tools are used internationally, references discussing cross-cultural or linguistic adaptations of adherence measures in cancer care would be useful for emphasizing the need for validation across diverse populations.
• In light of these comments authors may add these references:
o McDonald HP, Garg AX, Haynes RB. Interventions to Enhance Patient Adherence to Medication Prescriptions: Scientific Review. JAMA. 2002;288(22):2868–2879.
o Lam WY, Fresco P. Medication Adherence Measures: An Overview. Biomed Res Int. 2015: 217047.
o Haynes, R.B., et al. "Interventions for helping patients to follow prescriptions for medications." Cochrane Database of Systematic Reviews, 2002.
o Reeve, B.B., et al. "ISOQOL recommends minimum standards for patient-reported outcome measures used in patient-centered outcomes and comparative effectiveness research." Quality of Life Research, 2013.
o Herdman, M., Fox-Rushby, J., Badia, X. "Equivalence and the translation and adaptation of health-related quality of life questionnaires." Quality of Life Research, 1997.

Figures & Tables
The quality the tables supports the clarity of the results. However, reviewer thinks that Table 3 that talks about the methodological quality assessment may be rearranged in an alphabetical order to be easily read by the readers.

Experimental design

Material and Methods
The methods used in the study correlate well with study’s aim but reviewer thinks authors may
clarify the criteria used to include tools specific to OAAs and adult cancer populations while excluding studies on other medications (e.g., analgesics) or non-cancer groups. While describing the method author may mention cross-cultural and linguistic adaptation of adherence measures to emphasize the need for validation across diverse populations, which would add to study's applicability. Also, noting that PRISMA guidelines were used for systematic reporting of search results (including databases searched, inclusion/exclusion process, and final study selection) helps establish the study’s thoroughness. PRISMA is a widely respected standard in systematic reviews, and referencing its application would add credibility to the review’s findings and methodology.

Validity of the findings

Results
The study provides a comprehensive comparison of adherence tools for OAAs, offering valuable insights into their psychometric properties. The results are novel. In order to add more insights to the results, authors may expand on the limitations of widely-used tools for OAA adherence to underscore the unique adherence challenges in oncology. Also, the potential impact of self-reported adherence limitations and recommendations for developing more objective adherence metrics may be mentioned to reinforce the study’s relevance.

Discussion
The findings well correlate with results however authors may acknowledge the retrospective limitations and discuss potential future studies that may improve adherence measurement in cancer treatment.

Conclusion
The conclusions align well with the study's findings, emphasizing the need for reliable adherence measures specific to OAAs and the limitations of non-specific tools. The focus on patient stratification and the application of validated tools in cancer treatment is well-supported by the results. Authors may recommend the criteria for selecting adherence tools based on their psychometric properties and relevance to cancer populations.

Additional comments

no comment

Reviewer 2 ·

Basic reporting

Thank you for your time and effort in conducting this systematic review titled "Psychometric properties of instruments for assessing adherence to oral antineoplastic agents: a COSMIN systematic review", which was a pleasure to read.

I commend the authors for following closely to both the COSMIN and PRISMA guideline. I would like to suggest the following amendments in the Tables to improve clarity.

Table 1:
- to use "etc" instead of "et al" in cancer diagnosis column
- suggest to list the mean age of study participants and PROMs response rate for each study, if applicable

Table 3:
- consider to replace alphabet "V,A,D" to numbering, for example, 4=very good, 3=adequate, 2=doubtful, 1=inadequate, so that reader may have a quick aggregated overall viewpoint on the quality of item assessed.
- suggest to add the PROM name like in Table 3 and Table 4.
- add in table footnote, details of how the rating was determined, i.e.: "To determine the overall rating of the quality of each single study on a measurement property, the lowest rating of any standard in the box is taken (i.e. “the worst score counts” principle)"

Table 4:
- suggested title: The quality of psychometric property measurement and evidence
- suggested "Recommendation" column title: Recommendation for use to assess adherence to OAA
- since the symbol "B" was not explained in footnote, suggest to input "weakly recommended" in the Recommendation column, concurrent with the body of the manuscript.
- add in table footnote, details of how the levels of recommendation is determined, e.g. strongly recommended=PROMs with evidence for sufficient content validity (any level) AND at least low quality evidence for sufficient internal consistency
-standardize each first word as capital letters (currently, some are written as capital letters, some are written as small letter)

Experimental design

Line 55-61:
Clarify what are the differences in OAA compared to other disease medication (eg antihypertensives) that justifies its own adherence scale

Line 81-82:
Clarify what are the advantages of COSMIN compared to the evaluation tool used by Claros et al. 2019

Line 123:
I would like to suggest to include a supplementary data/note on how papers in non-English languages obtained were managed. Were the papers translated and how, and/or were all reviewers fluent in all languages to be able to synthesis the data?

Line 104-105:
How were the search for the reference list conducted? (e.g. if manually searched, kindly state).

Line 112:
Since analgesics and antiemetics are not OAA (inclusion criteria), suggest to rephrase to "(1) studies that included mainly oral agents other than OAA (such as analgesics and antiemetics)"

Line 113-114:
Suggest to add: (2) studies in which a PROM was "solely" used to measure an outcome (e.g., randomized clinical trials)

Line 114:
Kindly clarify "(4) review"

Line 123:
How was the management of unclear/missing data? e.g: were study investigators contacted?

Validity of the findings

Line 318:
Suggest to add: However, none met the COSMIN criteria, and all the PROMs were weakly recommended "for use to assess OAA adherence"

Line 270-272 and Line 319:
In discussion, it is mentioned both T-OCAS and EXPAD-ANEO met the highest number of measurement properties. In the conclusion, EXPAD-ANEO is provisionally recommended. Kindly close this gap in the discussion, i.e. the disadvantages of T-OCAS compared to EXPAD-ANEO, which would provide justification of your conclusion regarding EXPAD-ANEO.

Line 322-323:
One of the conclusion stated "We suggest using the COSMIN methodology for developing, adapting, and validating OAAs adherence scales in future research.". Although this was an interesting observation, it was not stated explicitly as an objective. May I suggest that the basis of suggesting the use of the COSMIN methodology also be added to the discussion.

Additional comments

In general, this paper is clearly written and well organized. Some minor grammar/typo correction, and sentence structure amendment to improve clarity, are suggested as follows (kindly refer to the PDF version for line number):

Line 17 to unbold the full-stop

Line 18 to add:
.. to examine "the quality of" their...

Line 34 and Line 320-321 to add:
...and high quality for "sufficient" structural validity...

Line 55-58 too lengthy and suggest to break-up:
Most studies used the Morisky Medication Adherence Scale (MMAS-8) [13] or its modified form to assess OAA adherence. Originally developed for patients with hypertension, MMAS-8 applicability and effectiveness in cancer patients treated with OAAs remains to be verified.

Line 63 to remove comma after "pharmacy refill rates", and to add
comma after "Medication Event Monitoring System (MEMSCap™), "

Line 66, since the author does not appear to be referring to any specific PROMs, to omit "these":
..."In contrast, PROMs are easy to use"...

Line 84 to add:
...reported "the quality of" their psychometric properties...

Line 86-90:
The last point is lengthy, thus it is suggested to re-phrase this sentence.

Line 93 to substitute "psychometric properties" to:
"PROMs"

Line 93 to add:
...and "was" reported following the the Preferred Reporting ...

Line 118-122 suggest to amend to:
Two "reviewers" (Miaomiao Sun and Kanghui Huang)"," trained in evidence-based methodologies"," independently screened the titles and abstracts of the references "according to the predefined eligibility criteria," and then retrieved the full text of articles that may "have" wholly or potentially "met" the eligibility criteria.

Line 126 to amend to:
...A standardized extraction table was "utilized, which included" the ...

Line 137 to add:
...The checklist comprises "of" ten boxes...

Line 140 to add:
... The "methodological quality of" items in each box "was" rated as...

Line 141 to add:
...The quality of the "criteria determining good" measurement properties...

Line 166 to add:
...Table 1 presents detailed information on the studies that "were" included.

Line 177 missing full stop at the end of the sentence.

Line 183 omit "Moreover".

Line 194 to add:
...All instruments were "either" self-reported...

Line 219:
Did the authors mean "conditions" instead of "conditional"?

Line 227 to add:
...Table 4 displays the "quality of" psychometric properties and evidence...

Line 239 to amend to:
...one PROM (Baudot et al., 2016) was downgraded to very low due to "an" extremely...

Line 243 to amend to:
...Cronbach's alpha for each subscale "were not reported".

Line 268-269 to amend to:
This review identified among adult cancer patients, eight PROMs to assess adherence to OAAs, which included endocrine, cytotoxic, targeted therapies, and immunomodulators.

Line 270 and Line 318 to add:
...and all the PROMs were weakly recommended "for use to assess adherence to OAAs."

Line 274-275 to amend to:
According to the COSMIN guidelines [17], content validity is regarded as "the" most critical psychometric "property of PROMs".

Line 278 to add comma:
...with chronic diseases to measure medication adherence","

---

## Round 0.2 · Minor Revisions

Dear Authors,

I am happy to let you know that the manuscript is suitable for publication, pending the remaining minor revisions suggested by Reviewer#2 to improve the clarity of the manuscript.

Reviewer 1 ·

Basic reporting

Title & Abstract
The title accurately reflects the manuscript's content. The abstract summarizes the research purpose, methodology, and key findings effectively, with updates emphasizing the COSMIN framework and the limitations of existing tools. The authors made no additions about alternative adherence frameworks as per their response, but the abstract remains clear and aligns with the focus of the study.

Introduction/Background
The authors clarified the unique adherence challenges of oral antineoplastic agents (OAAs), including side effects, cost, and regimen complexity, differentiating them from other medications. They incorporated suggestions about the importance of OAA-specific tools, supported by updated references. Limitations of non-specific tools like the MMAS-8 were addressed, adding robustness to the theoretical foundation. Additional information about content validity and cross-cultural adaptation strengthens the rationale for the study.

Figures and the Tables
The revised Table 3 has been updated and reorganized to address the reviewers’ suggestions. PROM names have been included in the table to facilitate easier identification and cross-referencing with other sections of the manuscript. Additionally, a footnote has been added to explain the rating system, particularly the "worst score counts" principle, providing clarity on how the ratings were determined. While the reviewers recommended using numeric ratings (1-4) for simplicity, the authors retained the alphabetic system (V, A, D), citing its familiarity in COSMIN systematic reviews. The reorganization of the table enhances its overall readability and accessibility.

Experimental design

Materials and Methods
The methods section was expanded to include detailed inclusion and exclusion criteria, reflecting the study's focus on OAA-specific tools and adult cancer populations. The authors highlighted the use of PRISMA guidelines for systematic reporting and included discussion on cross-cultural adaptations of tools, emphasizing the validation across diverse populations. The management of non-English articles and the handling of unclear data were also discussed, addressing reviewers' concerns comprehensively.

Validity of the findings

Results
The results section provides a comprehensive comparison of tools. Limitations of widely used tools were elaborated, emphasizing unique adherence challenges in oncology. The authors highlighted the inadequacies in psychometric properties across tools and explained why certain instruments were weakly recommended. Tables were updated for clarity and organized systematically, although the authors opted to retain the COSMIN notation system ("V, A, D") to align with standard practices.

Discussion
The discussion effectively correlates findings with results. The limitations of subjective tools and the need for objective adherence metrics were noted. The authors added recommendations for improving measurement properties and validated instruments for diverse cancer populations. They also emphasized future research directions, including refining methodologies for PROMs and adapting tools for broader applications.

Conclusion
The conclusions align well with the study findings, highlighting the limitations of existing tools and the need for instruments with stronger psychometric properties. The authors revised statements about instrument recommendations, emphasizing the dependence on psychometric properties and tool relevance.

Reviewer 2 ·

Basic reporting

no comment

Experimental design

Line 131:
Omit “(2) participants were non-cancer populations;”.
Rationale: Inclusion criteria already specified “diagnosed with cancer”

Line 140 (2.4 Data Extraction):
Please specify if original authors were contacted for missing or unreported data.

Line 282:
“one PROM (Qin et al., 2020) was downgraded to low due to a serious risk of bias and a sample size of less than 100,”
Was sample size already considered in assessing the COSMIN RoB? Consider the COSMIN guideline: imprecision principle should not be used for measurement properties in which a sample size requirement is already included in the COSMIN Risk of Bias box, i.e. content validity, structural validity, and cross‐cultural validity.

Validity of the findings

Line 236:
“or if each was tested in an inappropriate number of patients”:
Does the author mean that the sample size was small? If so, then suggest to maintain the original sentence and to omit “if”.
The entire sentence could be restructured as such: “The quality of three PROM development studies (Baudot et al., 2016; Gambalunga et al., 2022; Talens et al., 2023) was rated doubtful because each item was tested in an inappropriate number of patients or it was unclear if all items were tested in their final form.”

Line 329-336:
Suggested to move to Results section.

Line 389:
“Finally, patient-reported adherence rates were higher than those of objective measures across studies (Atkinson et al., 2016). Thus, subjective instruments are no substitute for objective measurements.”
Please clarify these two sentences.

Additional comments

Some sentence structure amendment to improve clarity, are suggested as follows (kindly refer to the PDF version for line number):

Line 28:
“Eight studies from eight identified instruments were eventually included.”
Suggest to amend: “Eight studies assessing eight identified instruments were included”.

Line 30:
“However, none of the instruments explored measurement error, cross-cultural validity/measurement invariance, and responsiveness.”
Suggest to amend: “None of the studies explored measurement error, cross-cultural validity/measurement invariance, and responsiveness of the instruments.”

Line 32-34
“Eventually, one instrument was not recommended for assessing OAA adherence due to high-quality evidence for insufficient internal consistency, and seven were only weakly recommended for use to assess OAA adherence.”
Suggest to reorder from strongest to weakest recommendation (example as per written in Line 323)

Line 35:
“The selection of the most appropriate instrument depends on its psychometric properties and relevance to the type of OAA.
Suggest to amend: “The selection of the most appropriate instrument to assess adherence to OAA depends on its psychometric properties and relevance to the type of OAA.”

Line 39:
“…OAA adherence in cancer patients in the future.”
Suggest to amend: “…OAA adherence in cancer patients.”

Line 56:
“Ensuring medication adherence is the key to ensuring treatment outcomes...”
Suggest to amend: “Ensuring medication adherence is the key to ensuring optimal treatment outcomes...”

Line 60:
“Objective measures such as pill counts, medication possession ratio, pharmacy refill rates, and the Medication Event Monitoring System (MEMSCap™). Subjective measures contain various …”
Suggest to amend: “Objective measures include pill counts, medication possession ratio, pharmacy refill rates, and the Medication Event Monitoring System (MEMSCap™), whereas subjective measures consist of various ...”

Line 67:
“A systematic review showed that rates for adherence to OAAs …”
Suggest to amend: “A systematic review reported that rates for adherence to OAAs …”

Line 67:
“0riginally developed for patients with hypertension, MMAS-8…”
Suggest to amend: “Originally developed for patients with hypertension, the MMAS-8…”

Line 74:
“Secondly, OAAs tend to be more expensive and may stop cancer patients from taking them due to financial considerations (Neugut et al., 2011).”
Suggest to amend: “Secondly, OAAs tend to be more expensive, which may deter cancer patients from adhering to treatment due to financial constrains (Neugut et al., 2011).”

Line 76:
“Additionally, chemotherapy regimens can be quite complex, requiring patients to take between 5 to 12 pills two to three times daily, sometimes following confusing schedules like...”
Suggest to amend: “Additionally, chemotherapy regimens can be quite complex, for example, having to consume 5 to 12 pills two to three times daily, and may involve confusing schedules like…”

Line 79:
“However, the above-discussed factors are not highlighted in the…”
Suggest to amend: “However, the above-discussed factors are not emphasized in the…”

Line 150:
“Two reviewers (MMS, KHH) independently appraised methodological quality and measurement properties.”
Suggest to amend: “Two reviewers (MMS, KHH) independently appraised the methodological quality and measurement properties of the PROMs.”

Line 175:
“…sufficient internal consistency that can be categorized as “strongly recommended”;”
Suggest to omit “that”

Line 176:
“…insufficient psychometric property that should be..”
Suggest to omit “that”

Line 237:
“For content validity, five studies (Bagcivan & Akbayrak, 2015; Baudot et al., 2016; Gambalunga et al., 2022; Li, Sun & Dong, 2018; Talens et al., 2023) evaluated the relevance of the PROM items, four studies (Bagcivan & Akbayrak, 2015; Baudot et al., 2016; Gambalunga et al., 2022; Talens et al., 2023) evaluated comprehensibility of the PROM items and only two studies (Baudot et al., 2016; Talens et al., 2023) evaluated comprehensiveness of the PROM items, all of which were rated adequate (Bagcivan & Akbayrak, 2015; Gambalunga et al., 2022) or doubtful (Baudot et al., 2016; Li, Sun & Dong, 2018; Talens et al., 2023) due to unclear whether professionals from all required disciplines were included or each item was tested in an inappropriate number of patients or professionals.”
Suggestion: Thank you for the detailed explanation to justify the rating. However, the sentence is too lengthy, and suggest to break.

Line 252:
“were rated doubtful due to whether the scale was unidimensional.”
Suggest to amend: “were rated doubtful due to uncertainty if the scale was unidimensional.”

Line 255:
“…internal subscale, and only the correlation coefficient between entries was calculated.”
Suggest that if not all studies exhibit both weaknesses, amend to the original text: “…internal subscale, or only the correlation coefficient between entries was calculated.”

Line 256:
“Four studies (Bagcivan & Akbayrak, 2015; Gambalunga et al., 2022; Li, Sun & Dong, 2018; Qin et al., 2020) reported reliability, but none calculated the intraclass correlation coefficient (ICC) or specified whether the patients remained stable, the conditions were comparable, or if the time interval was suitable, making the reliability doubtful.”
Suggest to amend: “Four studies (Bagcivan & Akbayrak, 2015; Gambalunga et al., 2022; Li, Sun & Dong, 2018; Qin et al., 2020) reported reliability, but none calculated the intraclass correlation coefficient (ICC). Furthermore, none of the four studies specified whether the patients remained stable, if the conditions were comparable, or if the time interval was suitable, making the reliability doubtful.”

Line 262:
“Besides, hypothesis testing for construct validity assessed by three studies…”
Suggest to omit “Besides”, as this sentence is not related to the previous sentence.

Line 280:
“The quality of evidence for structural validity of one PROM (Talens et al., 2023) was high, two PROMs (Bagcivan & Akbayrak, 2015; Li, Sun & Dong, 2018) were downgraded to moderate quality evidence due to a serious risk of bias…”
Suggest to amend: “The quality of evidence for structural validity of one PROM (Talens et al., 2023) was high, whereas two PROMs (Bagcivan & Akbayrak, 2015; Li, Sun & Dong, 2018) were downgraded to moderate quality of evidence due to a serious risk of bias…”

Line 322:
“However, none met the COSMIN criteria”
Suggest to amend: “However, none met the COSMIN criteria, and at best were only weakly recommended for assessing OAA adherence. In addition, none of the PROMs explored measurement error, cross-cultural validity/measurement invariance, and responsiveness.”

Line 342:
“MMAS-8 (Qin et al., 2020), TAM (Silveira et al., 2021),…”
Suggest to amend: “The other three: MMAS-8 (Qin et al., 2020), TAM (Silveira et al., 2021) and the self-assessment adherence questionnaire (Daouphars et al., 2013), were not originally developed for cancer patients.”

Line 355:
“All studies evaluated internal consistency, but only two had excellent methodological quality. It is crucial to calculate Cronbach’s alpha or omega for each subscale.”
Suggest to amend: “All studies evaluated internal consistency, but majority had doubtful or inadequate methodological quality. Therefore, it is crucial to determine the Cronbach’s alpha or omega for each subscale, compared to an internal consistency statistic for the overall tool.”

Line 399
Instruments with rigid measurement properties are urgently needed to be developed to assess OAA adherence in cancer patients in the future.
Suggest to omit “in the future”.

Annotated reviews are not available for download in order to protect the identity of reviewers who chose to remain anonymous.

---

## Round 0.3 · accepted · Accept

I am glad to inform you that the manuscript is now suitable for publication on PeerJ.